# Reducing the Impact of Headache and Allodynia Score in Chronic Migraine: An Exploratory Analysis from the Real-World Effectiveness of Anti-CGRP Monoclonal Antibodies Compared to Onabotulinum Toxin A (RAMO) Study

**DOI:** 10.3390/toxins16040178

**Published:** 2024-04-07

**Authors:** Danilo Antonio Montisano, Riccardo Giossi, Mattia Canella, Claudia Altamura, Marilena Marcosano, Fabrizio Vernieri, Alberto Raggi, Licia Grazzi

**Affiliations:** 1Neuroalgology Unit and Headache Center, Fondazione IRCCS Istituto Neurologico Carlo Besta, Via Celoria, 11, 20133 Milan, Italy; 2Poison Control Center and Clinical Pharmacology Unit, Grande Ospedale Metropolitano Niguarda, Piazza Ospedale Maggiore 3, 20162 Milan, Italy; 3Department of Research and Clinical Development, Fondazione IRCCS Istituto Neurologico Carlo Besta, Via Celoria, 11, 20133 Milan, Italy; 4Department of Medical Biotechnology and Translational Medicine, Postgraduate School of Clinical Pharmacology and Toxicology, Università degli Studi di Milano, Via Vanvitelli, 32, 20129 Milan, Italy; 5Neuroimmunology and Neuromuscular Disease Unit, Fondazione IRCCS Istituto Neurologico Carlo Besta, Via Celoria 11, 20133 Milan, Italy; 6Fondazione Policlinico Universitario Campus Bio-Medico, Via Alvaro del Portillo, 200, 00128 Roma, Italy; 7Department of Medicine and Surgery, Università Campus Bio-Medico di Roma, Via Alvaro del Portillo, 21, 00128 Roma, Italy; 8Neurology, Public Health and Disability Unit, Fondazione IRCCS Istituto Neurologico Carlo Besta, 20131 Milan, Italy; 9SC Neuroalgologia–Centro Cefalee, Fondazione IRCCS Istituto Neurologico Carlo Besta, Via Celoria 11, 20133 Milan, Italy

**Keywords:** migraine, erenumab, galcanezumab, fremanezumab, onabotulinum toxin A, allodynia, HIT-6, ASC-12

## Abstract

Background: Chronic migraine (CM) is a disabling and hard-to-treat condition, associated with high disability and high cost. Among the preventive treatments, botulinum toxin A (BoNT-a) and monoclonal antibodies against the calcitonin gene-related protein (anti-CGRP mAbs) are the only disease-specific ones. The assessment of the disease burden is complex, and among others, tools such as the allodynia symptoms checklist (ASC-12) and headache impact test (HIT-6) are very useful. This exploratory study analysed the impact of these two therapies on migraine burden. Methods: The RAMO study was a multicentre, observational, retrospective investigation conducted in two headache centres: the Fondazione IRCCS Istituto Neurologico Carlo Besta (Milan) and the Fondazione Policlinico Campus Bio-Medico (Rome). This study involved patients with chronic migraine treated with mAbs or BoNT-A. We conducted a subgroup exploratory analysis on HIT-6 and ASC-12 scores in the two groups. The Wilcoxon rank-sum test, Fisher’s exact test, and ANOVA were performed. Results: Of 126 patients, 36 on mAbs and 90 on BoNT-A had at least one available follow-up. mAbs resulted in a mean reduction of −11.1 and −11.4 points, respectively, in the HIT-6 at 6 and 12 months, while BoNT-A was reduced −3.2 and −3.6 points, respectively; the mAbs arm resulted in mean reductions in ASC-12 at 6 and 12 months of follow-up of −5.2 and −6.0 points, respectively, while BoNT-A showed lesser mean changes of −0.5 and −0.9 points, respectively. The adjusted analysis confirmed our results. Conclusions: In this exploratory analysis, anti-CGRP mAbs showed superior effectiveness for HIT-6 and ASC12 compared to BoNT-A. Reductions in terms of month headache days (MHD), migraine disability assessment test (MIDAS), and migraine acute medications (MAM) were clinically relevant for both treatments.

## 1. Introduction

Among the types of primary headache, chronic migraine (CM) is a condition characterized by a headache frequency of 15 or more headache days/month, of which at least 8 per month present typical migraine features, for at least 3 consecutive months, according to the third International Classification of Headache Disorders (ICHD-3) of the International Headache Society (IHS) [1]. It represents the evolution of episodic migraine (EM) for about 5% of all patients with migraine. The risk factors for this evolution are high headache frequency, inadequate acute migraine treatment, high headache-related disability, obesity, anxiety, allodynia, and depression [2].

According to the Global Burden of Disease (GBD), the disability burden of migraine is among the top overall causes of disability and in CM, it is even more elevated, especially when associated with medication overuse headache (MOH), leading to reduced quality of life (QoL) and a considerable impact on patients and society [3,4,5]. The costs of this condition are also important, as CM has a much higher cost than EM [6]. So, the accurate treatment of CM and CM with MOH and their reverting to EM produces a relevant cost-sparing in terms of direct and indirect costs [7].

Migraine treatment is divided into acute and preventive treatments. The aim of preventive treatment is to decrease the frequency of headaches, mitigate the severity of attacks, reduce patient disability, and minimize the risk of medication overuse. If preventive treatments are successful, they can additionally enhance patients’ QoL and improve their response to acute treatments [8,9,10].

For several years, the preventive treatment of migraine used non-specific drugs, originally designed for different purposes, frequently linked to suboptimal and unpredictable effectiveness, as well as challenges in terms of tolerability and adherence [2].

The history of migraine preventive treatment changed when in 2010 the Food and Drug Administration (FDA) approved the use of the first migraine-specific treatment, onabotulinum toxin A (BoNT-A), based on the findings of the Phase III Research Evaluating Migraine Prophylaxis Therapy (PREEMPT) trial [11,12]. BoNT-A represented the only CM-approved treatment for several years, with good data on safety, tolerability, and efficacy for clinical and disability outcomes [13,14,15,16,17].

Another milestone came when monoclonal antibodies against the calcitonin gene-related protein (anti-CGRP mAbs) were approved as preventive treatments [18]. Anti-CGRP mAbs are effective and well-tolerated in CM, even for the refractory form [8,19,20,21,22,23,24].

Apart from the head-to-head study of erenumab against topiramate (HER-MES study), which demonstrated a higher efficacy and safety of erenumab when compared to topiramate, direct comparisons between CM preventive treatments are lacking [25]. Recently, our group published the results of the Real-world effectiveness of Anti-CGRP Monoclonal antibodies compared to Onabotulinum toxin A (RAMO) study, which showed a superior effectiveness of anti-CGRP mAbs when compared to BoNT-A, with comparable safety profiles [26].

The definition of CM burden is not simple, and there is no global consensus about which parameters to consider when defining it (frequency, disability, clinical indexes, quality of life, workplace productivity, or costs) [27]. To describe the clinical burden in this population, two assessment tools have been used in research and clinical practice that well describe the impact of the disease on patients: the headache impact test (HIT-6) and the allodynia symptoms checklist (ASC-12). HIT-6 is a comprehensive metric of the negative impact of headaches that has been developed for use in the screening and tracking of headache patients in either clinical practice or clinical research, as well as episodic and chronic migraine, with a good reliability for the intensity of the attack [28,29,30].

ASC-12 evaluates allodynia, which is related to migraine severity as it represents an indirect sign of the central sensitization process involved in CM development [31,32] and has even been suggested as a predictive factor of the treatment response [33,34,35].

In this paper, we present an exploratory analysis of the RAMO study on the effectiveness of anti-CGRP mAbs and BoNT-A on HIT-6 and ASC-12.

## 2. Results

### 2.1. Study Population Characteristics

Up to 1 October 2023, from a total of 216 potentially eligible patients, 183 met the inclusion criteria for the RAMO study, 86 on anti-CGRP mAbs and 97 on BoNT-A. Of them, 126 patients, 36 on anti-CGRP mAbs and 90 on BoNT-A, had at least one available follow-up for HIT-6 or ASC-12 for inclusion in this subgroup exploratory analysis. The mean age of the overall population was 46.3 years with 28.8 years of mean migraine history duration. Females were more represented (86.5%) with a significantly higher presence in the BoNT-A arm (*p* = 0.023). Non-significant differences were observed for age, migraine duration, MOH, and presence of tension-like symptoms (TLS) (Table 1). Patients included in the BoNT-A arm had superior monthly headache days (MHD) (*p* = 0.0116) at baseline and presented more comorbidities (*p* < 0.001) and concomitant migraine medication use (*p* < 0.001). Non-significant differences in between arms were observed for baseline migraine disability assessment test (MIDAS), migraine acute medications (MAM), and HIT-6 score, while patients in the anti-CGRP mAbs arm presented a superior mean ASC-12 score (*p* = 0.0005) (Table 1).

### 2.2. Primary Efficacy Outcomes

Anti-CGRP mAbs resulted in significantly reduced mean HIT-6 and ASC-12, both at 6 and 12 months of follow-up, compared to BoNT-A. In particular, anti-CGRP mAbs resulted in a mean change from baseline of −11.1 and −11.4 points in the HIT-6 at 6 and 12 months, respectively, while the BoNT-A mean change from baseline was −3.2 and −3.6 points (*p* < 0.0001 and *p* = 0.0042, respectively). The adjusted analysis yielded similar significant results at 6 (MD −10.7 points; 95% CI from −18.9 to −2.7; *p* = 0.010) and at 12 months (MD −13.4; 95% CI from −23.4 to −3.5; *p* = 0.009) of follow-up (Table 2).

The anti-CGRP mAbs arm resulted in a significantly greater mean change from baseline in the ASC-12 score at 6 and 12 months of follow-up (−5.2 and −6.0 points, respectively), while BoNT-A showed lesser mean changes from baseline (−0.5 and −0.9 points, respectively) (*p* = 0.0056 and *p* = 0.0011, respectively). The adjusted analysis substantially replicated the results at 6 (MD −5.3 points; 95% CI from −9.2 to −1.5; *p* = 0.008) and 12 months (MD −4.4 points; 95% CI from −8.0 to −0.9; *p* = 0.016) of follow-up (Table 2). The sensitivity analysis performed by stratifying patients on the basis of the minimal baseline ASC-12 cutoffs yielded comparable results (Appendix A).

### 2.3. Other Efficacy Outcomes

For all other efficacy outcomes, anti-CGRP mAbs resulted in significantly greater reductions for MHD, MIDAS, and MAM, both at 6 and 12 months of follow-up, compared to BoNT-A. Full descriptions of the outcome changes from baseline are described with *p*-values in Table 3. When adjusting for baseline confounders, the MHD results substantially replicated the unadjusted results with anti-CGRP mAbs, showing a significant reduction compared to BoNT-A at 6 (MD −8.9 days; 95% CI from −16.2 to −1.7; *p* = 0.017) and 12 months (−7.9 days; 95% CI from −14.8 to −0.9; *p* = 0.027) of follow-up. Differently, MIDAS resulted in a non-significantly difference between the two treatment arms at 6 (MD −36.3 points; 95% CI from −97.2 to 24.6; *p* = 0.238) and 12 months (MD −33.1 points; 95% CI from −89.4 to 23.3; *p* = 0.244). The MAM reduction showed a non-significant difference at 6 months (MD −10.9 pills; 95% CI from −22.2 to 0.5; *p* = 0.061), while at 12 months, the anti-CGRP mAbs resulted in a significantly different MAM reduction (MD −10.3 pills; 95% CI from −19.4 to −1.2; *p* = 0.028) compared to BoNT-A (Table 3).

## 3. Discussion

The results of this exploratory analysis of the RAMO study has shown a meaningful divergence in the decrease of HIT-6 and ASC-12 between the two treatment groups, with anti-CGRP mAbs showing superior reductions in both outcomes at the evaluated time points compared to BoNT-A.

In our study group, patients treated with BoNT-A had higher baseline MHD, more comorbidities, and more frequently consumed concomitant medications, so they had some features suggestive of a more severe migraine situation than patients receiving anti-CGRP mAbs. To mitigate the influence of these potentially significant confounding elements, we have carried out an adjusted analysis, which nevertheless basically confirmed the results achieved from the unadjusted analysis.

The anti-CGRP mAbs group, which included all three mAbs available in the investigated study period, reported a significant mean reduction for HIT-6 scores of −11.1 and −11.4 points at 6 and 12 months, respectively, slightly inferior to what was observed in a prospective cohort study on the same drugs, which reported a mean improvement of −14.53 and −16.09 points at 6 months and 12 months, respectively [36]. Conversely and similar to our findings, another observational study on 152 patients with high-frequency episodic migraine and CM showed a mean reduction of −11.7 points at 6 months of follow-up [37]. The BoNT-A group showed a smaller reduction of HIT-6 score at 6 and 12 months compared to anti-CGRP mAbs, which, however, was at the limit of clinical relevance. Our data conflicts with other experiences in the literature that showed a slightly superior mean reduction of HIT-6, being at least 5 points or even more [38,39].

In the anti-CGRP mAbs arm, we observed a mean ASC-12 reduction of −5.2 and −6.0 points at 6 and 12 months, respectively, differently from what was previously reported in a prospective cohort study including 70 patients with CM who received erenumab, where the mean ASC-12 reduction resulted in about −1.9 points at 6 months of follow-up. However, the patients included in that study had an inferior baseline ASC-12 score, which may limit the comparability to our results [40]. Nevertheless, the BoNT-A arm resulted in mean reductions at lower margins of clinical relevance for HIT-6 and did not show a clinically relevant mean reduction for ASC-12. Indeed, the lower baseline ASC-12 in the BoNT-A group compared to anti-CGRP mAbs could have limited the available improvement for this outcome, even after the confirmation of the results by our sensitivity analysis stratified by ASC-12 baseline score, since it led to a considerable reduction in the sample size.

Our results on ASC-12 parallel those of a previous work on BoNT-A of our group, whose population included the subset of BoNT-A patients of the RAMO study from the Milan centre [41]. Conversely, other studies showed that BoNT-A in CM can be effective by influencing central sensitisation, by modulating nociceptive drive, and by reducing the overall burden of migraine [32], and the ASC-12 score decreased significantly in a similar treated population [42,43].

Apart from the main results of the RAMO retrospective study, randomized controlled trials directly comparing anti-CGRP mAbs and BoNT-A for migraine are not available, so far [26]. A previous meta-analysis on placebo-controlled randomized trials indirectly comparing anti-CGRP mAbs to BoNT-A has shown similar efficacy of the two treatments in terms of migraine days, MHD, 50% response rate, MAM, and HIT-6 with a reduced incidence of adverse events in the anti-CGRP mAbs group [44]. Possible reasons for these discrepancies from our results are extensively discussed in our previous publication. However, for the HIT-6 results, the role of a contextual effect on patients receiving anti-CGRP mAbs and residual bias due to the superior migraine severity of patients in the BoNT-A arm could not be excluded, even after the adjusted analysis [26,45].

Eventually, in this subgroup analysis, the MHD, MIDAS, and MAM changes from baseline substantially replicated the results of the full-set analysis of the RAMO study [26], so even with a difference in the reduction of HIT-6 and ASC-12 scores, both treatments reached a suitable effectiveness in clinical outcomes, with anti-CGRP mAbs possibly resulting in superior efficacy compared to BoNT-a. However, these results are far from conclusive, and future larger studies should be conducted.

The burden of migraine is complex, and its multifaceted nature can make it challenging to fully understand it. Indeed, the majority of the studies focus on only a few specific domains, such as those covered in tools like the MIDAS, and they may not capture the full range of challenges faced by patients with migraine. As a result, our understanding of the migraine burden from a patients’ perspective remains limited [46]. The HIT-6 offers a quite unique insight into the impact of migraine treatments on patients’ daily lives, and qualitative evidence shows that each item on the HIT-6 rates a concept highly pertinent to migraine patients [47]. Cutaneous allodynia is a very common feature in this condition, reflecting the presence of central sensitization, and it is also associated with the progression of migraine and may increase the likelihood of an inadequate response to acute treatment. Thus, allodynia assessment with ASC-12 is an important tool in clinical practice [48]. Measures such as HIT-6 and ASC-12 are valuable allies, although not solely, in sharply defining the whole clinical burden of migraine patients and should be included in future studies.

### Limitations

The main limitation of this subgroup analysis of the RAMO study is the reduced number of patients with available data for HIT-6 and ASC-12 in the anti-CGRP mAbs arm. This is because these two outcomes are not systematically collected in our centres like MHD, MIDAS, and MAM, and some patients did not return the completed questionnaires during their outpatient visits. This could lead to potential selection bias since patients on anti-CGRP mAbs could more frequently have returned their questionnaire if they had higher allodynia. This could explain the significantly higher ASC-12 score at baseline among them. However, we produced an adjusted analysis accounting for this difference and performed a sensitivity analysis stratifying patients by their ASC-12 baseline severity, limiting potential bias in the effect estimate.

## 4. Conclusions

Similar to the primary study results of the RAMO study, in this exploratory analysis, anti-CGRP mAbs showed superior effectiveness as measured by the HIT-6 and ASC-12 compared to BoNT-A, even after statistical adjustment for potential baseline confounders. The reductions in terms of MHD, MIDAS, and MAM were clinically relevant for both treatments, with anti-CGRP mAbs resulting in significantly superior improvement. Our findings must be confirmed by further studies, with a more balanced and wider population.

## 5. Methods

### 5.1. Study Design and Population

The RAMO study was an observational, retrospective, multicentre study performed in two hospital centres in Italy: Fondazione IRCCS Istituto Neurologico Carlo Besta (Milan) and Fondazione Policlinico Campus Bio-Medico (Rome). Briefly, this study included patients with CM who had received either anti-CGRP mAbs or BoNT-A and had a baseline MIDAS score of 11 or more. The primary outcome was the variation from baseline in MHD at 12 months of follow-up. Other outcomes included the improvement from baseline in MIDAS and MAM. Outcome assessment was performed both at 6 and 12 months of follow-up. The full methods of the RAMO study are provided in the dedicated publication [26]. In participating centres’ clinical databases, data on HIT-6 and ASC-12 were available for a subset of patients with CM. For this reason, in the original design of the study we prespecified the collection of HIT-6 and ASC-12 at the same time points as the other study outcomes and we conducted a subgroup exploratory analysis on these patients. For the conduction of this study, we adhered to the STrengthening the Reporting of OBservational studies in Epidemiology (STROBE) guidelines.

### 5.2. Study Outcomes

The two primary outcomes of this subgroup analysis were the comparison between anti-CGRP mAbs and BoNT-A of HIT-6 and ASC-12 changes from baseline to 6 and 12 months of follow-up.

HIT-6 is a scale created to assess the negative impact of headache on normal activities. It is a self-reported outcome that evaluates the following items with a 5-point Likert scale (ranging from never to always) in the last 4 weeks: frequency of severe pain, frequency of limitation in the ability to do daily activities, frequency of wishing to lie down when having a headache, frequency of feeling too tired to work or do daily activities due to a headache, frequency of feeling fed up or irritated due to a headache, and frequency of limitations in concentrating on work or daily activities due to a headache [28].

The ASC-12 is a scale designed to evaluate the presence and the severity of allodynia. It is a self-reported outcome that evaluates the presence of increased pain or unpleasant sensations during the most severe headache when engaging in the following 12 items: wearing a necklace, wearing earrings, wearing glasses, wearing tight clothes, wearing a ponytail, wearing contact lenses, shaving the face, taking a shower, combing the hair, resting the head on a pillow, exposure to cold, and exposure to heat. Each item is scored up to a maximum of 3 points and allodynia is classified into 4 levels: (I) none (score 0–2); (II) mild (score 3–5); (III) moderate (score 6–8); and (IV) severe (score 9+) [49].

To allow a comparison with the main results of the RAMO study, we also reported the change from baseline in MHD, MIDAS, and MAM as secondary outcomes.

### 5.3. Statistical Analysis

The RAMO study was powered to detect a between-treatment difference of 3.2 days with a standard deviation (SD) of 7.9 days, a power of 0.8, and a significance level at 0.05 with a one-sided *t*-test, which would have been guaranteed by a sample of 170 patients. No formal sample size estimation was performed on secondary outcomes, including HIT-6 and ASC-12. Thus, all of the analyses included in this paper are to be interpreted as exploratory and the *p*-values are nominal.

Continuous variables are described as means (SD) or mean differences with 95% confidence intervals (95% CI). Categorical variables are described as counts and percentages. Unadjusted comparisons were performed with the Wilcoxon rank sum test, the Fisher’s exact test, or the chi-squared test, as applicable. Normality was assessed with the Shapiro–Wilk test. In order to adjust for baseline confounders, we implemented an analysis of variance (ANOVA) model. Marginal mean differences with 95% CI were estimated from the models. To further evaluate the ASC-12 results, a sensitivity analysis was performed stratifying patients on the basis of baseline ASC-12 scores with minimal cutoffs, i.e., mild (≥3 points), moderate (≥6 points), and severe (>9 points). No imputation for missing data was performed. We used STATA 15 (College Station, TX, USA: StataCorp LLC) software for all analyses.

## Figures and Tables

**Table 1 toxins-16-00178-t001:** Population baseline characteristics.

	Total (*n* = 126)	Anti-CGRP mAbs (*n* = 36)	BoNT-A (*n* = 90)	*p*-Value
Age, years, mean (SD)	46.3 (10.1)	46.6 (11.1)	46.1 (9.7)	0.8047
Sex, female, *n* (%)	109 (86.5)	27 (75.0)	82 (91.1)	0.023
Migraine duration, years, mean (SD)	28.8 (10.2)	28.4 (4.9)	29.0 (10.1)	0.7611
MOH, *n* (%)	101 (80.2)	28 (77.8)	73 (81.1)	0.805
TLS, *n* (%)	44 (34.9)	11 (30.6)	33 (36.7)	0.543
Anti-CGRP mAbs, *n* (%)				NA
Erenumab	11 (8.7)	11 (30.6)	-	
Fremanezumab	8 (6.4)	8 (22.2)	-	
Galcanezumab	17 (13.5)	17 (47.2)	-	
Centre, *n* (%)				0.009
Milan	76 (60.3)	15 (41.7)	61 (67.8)	
Rome	50 (39.7)	21 (58.3)	29 (32.2)	
Comorbidities, *n* (%)	69 (54.8)	10 (27.8)	59 (65.6)	<0.001
Hypertension, *n* (%)	16 (12.7)	2 (5.6)	14 (15.6)	0.151
Depression, *n* (%)	18 (14.3)	1 (2.8)	17 (18.9)	0.022
Anxiety, *n* (%)	33 (26.2)	2 (5.6)	31 (34.4)	0.001
Epilepsy, *n* (%)	0 (0.0)	0 (0.0)	0 (0.0)	NA
Cardiovascular, *n* (%)	5 (4.0)	0 (0.0)	5 (5.6)	0.320
Gastroenteric, *n* (%)	7 (5.6)	3 (8.3)	4 (4.4)	0.407
Chronic pain conditions, *n* (%)	3 (2.4)	1 (2.8)	2 (2.2)	1.000
Cancer, *n* (%)	2 (1.6)	0 (0.0)	2 (2.2)	1.000
Endocrine and metabolic, *n* (%)	5 (4.0)	1 (2.8)	4 (4.4)	1.000
Other, *n* (%)	18 (14.3)	5 (13.9)	13 (14.4)	1.000
Concomitant migraine medications, *n* (%)	63 (50.0)	1 (2.8)	62 (68.9)	<0.001
Beta-blockers, *n* (%)	16 (12.7)	1 (2.8)	15 (16.7)	0.039
TCA, *n* (%)	18 (14.3)	0 (0.0)	18 (20.0)	0.002
Anti-convulsant, *n* (%)	16 (12.7)	0 (0.0)	16 (17.8)	0.006
ARBs, *n* (%)	11 (8.7)	0 (0.0)	11 (12.2)	0.033
SSRI-SNRI, *n* (%)	19 (15.1)	0 (0.0)	19 (21.1)	0.002
Pizotifen, *n* (%)	2 (1.6)	0 (0.0)	2 (2.2)	1.000
SARI, *n* (%)	2 (1.6)	0 (0.0)	2 (2.2)	1.000
Flunarizine, *n* (%)	0 (0.0)	0 (0.0)	0 (0.0)	NA
MHD, days, mean (SD)	20.7 (6.3)	18.3 (4.9)	21.6 (6.5)	0.0116
MIDAS, points, mean (SD)	87.7 (60.4)	76.9 (40.0)	92.0 (66.6)	0.6269
MAM, administrations, mean (SD)	21.1 (10.2)	18.4 (6.1)	22.1 (11.3)	0.2402
HIT-6, points, mean (SD)	66.3 (6.0)	66.1 (4.0)	66.4 (6.7)	0.5275
ASC-12, points, mean (SD)	5.7 (5.1)	11.4 (5.2)	4.7 (4.4)	0.0005

Abbreviations: ASC-12 = 12 items allodynia symptoms checklist; ARBs = angiotensin receptor blockers; BoNT-A = onabotulinum toxin A; anti-CGRP = anti calcitonin gene-related protein; HIT-6 = headache impact test; mAbs = monoclonal antibodies; MAM = migraine acute medications; MHD = monthly headache days; MIDAS = migraine disability assessment test; MOH = medication overuse headache; SARI = serotonin antagonist and reuptake inhibitors; SD = standard deviation, SNRI = serotonin-norepinephrine reuptake inhibitors; SSRI = selective serotonin reuptake inhibitors; TCA = tricyclic antidepressants; TLS = tension-like symptoms. Among the comorbidities, Other includes allergic asthma, amenorrhea, atopic dermatitis, autoimmune hepatitis, autoimmune thyroid disease, chronic vein insufficiency, connectivitis, essential tremor, *Helicobacter pylori*, hepatitis B infection, hepatitis C infection, hip dysplasia, insomnia, osteoarthritis, pituitary adenoma, polycystic ovary, psoriasis, reduction surgery of the jaw, restless leg syndrome, scleroderma, tuberculosis test positivity, upper airways resistance syndrome, and urticaria.

**Table 2 toxins-16-00178-t002:** Efficacy outcomes.

	Anti-CGRP mAbs	BoNT-A	*p*-Value	Adjusted MD (95% CI)	*p*-Value
HIT-6					
6 months CFB, mean (SD)	−11.1 (7.2)	−3.2 (6.9)	<0.0001	−10.7 (−18.9 to −2.7)	0.010
	*n* = 35	*n* = 90			
12 months CFB, mean (SD)	−11.4 (12.2)	−3.6 (8.6)	0.0042	−13.4 (−23.4 to −3.5)	0.009
	*n* = 32	*n* = 49			
ASC-12					
6 months CFB, mean (SD)	−5.2 (5.8)	−0.5 (3.6)	0.0056	−5.3 (−9.2 to −1.5)	0.008
	*n* = 10	*n* = 61			
12 months CFB, mean (SD)	−6.0 (5.6)	−0.9 (3.3)	0.0011	−4.4 (−8.0 to −0.9)	0.016
	*n* = 10	*n* = 49			

Abbreviations: 95% CI = 95% confidence interval; ASC-12 = allodynia symptoms checklist; BoNT-A = onabotulinum toxin A; CFB = change from baseline; CGRP = calcitonin gene-related protein; HIT-6 = headache impact test; mAbs = monoclonal antibodies; MD = mean difference; SD = standard deviation. Unadjusted comparisons were performed by means of the Wilcoxon rank-sum test. Adjusted analysis was performed with an ANOVA model with significantly different baseline variables entered as covariates to estimate MD with 95% CI for continuous variables. For categorical variables, a logistic regression model with the same baseline covariates was used to estimate OR with 95% CI.

**Table 3 toxins-16-00178-t003:** Other efficacy outcomes.

	Anti-CGRP mAbs	BoNT-A	*p*-Value	Adjusted MD (95% CI)	*p*-Value
MHD					
6 months CFB, mean (SD)	−12.6 (5.6)	−7.2 (8.7)	0.0002	−8.9 (−16.2 to −1.7)	0.017
	*n* = 36	*n* = 90			
12 months CFB, mean (SD)	−12.1 (6.7)	−7.5 (8.5)	0.0065	−7.9 (−14.8 to −0.9)	0.027
	*n* = 34	*n* = 84			
MIDAS					
6 months CFB, mean (SD)	−60.0 (42.5)	−36.0 (60.4)	0.0038	−36.3 (−97.2 to 24.6)	0.238
	*n* = 36	*n* = 90			
12 months CFB, mean (SD)	−60.4 (42.3)	−43.5 (60.1)	0.0564	−33.1 (−89.4 to 23.3)	0.244
	*n* = 34	*n* = 78			
MAM					
6 months CFB, mean (SD)	−13.1 (6.4)	−6.6 (11.9)	0.0003	−10.9 (−22.2 to 0.5)	0.061
	*n* = 36	*n* = 90			
12 months CFB, mean (SD)	−12.1 (8.2)	−8.3 (11.0)	0.0141	−10.3 (−19.4 to −1.2)	0.028
	*n* = 34	*n* = 84			

Abbreviations: 95% CI = 95% confidence interval; ANOVA = analysis of variance; BoNT-A = onabotulinum toxin A; CGRP = calcitonin gene-related protein; mAbs = monoclonal antibodies; MAM = migraine acute medications, MD = mean difference; MHD = monthly headache days; MIDAS = migraine disability assessment test; SD = standard deviation. Unadjusted comparisons were performed by means of the Wilcoxon rank-sum test. Adjusted analysis was performed with an ANOVA model with significantly different baseline variables entered as covariates to estimate MD with 95% CI for continuous variables.

## Data Availability

The datasets used and/or analysed during the current study are available from the corresponding author on reasonable request.

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
