# Peer review of "Reducing the Impact of Headache and Allodynia Score in Chronic Migraine: An Exploratory Analysis from the Real-World Effectiveness of Anti-CGRP Monoclonal Antibodies Compared to Onabotulinum Toxin A (RAMO) Study"

_toxins, 2024, doi:10.3390/toxins16040178_

Round 1

Reviewer 1 Report

Comments and Suggestions for Authors

Manuscript entitled „Reducing the Impact of Headache and Allodynia score in Chronic Migraine: An Exploratory Analysis from the Real-World Effectiveness of Anti-CGRP Monoclonal Antibodies Compared to OnabotulinumtoxinA (RAMO) Study” is an interesting, well-written and well-planned an experimental work. However, the text needs some corrections according to the following comments:

Abstract

Line 9 – explain in full name an abbreviation CGRP

Line 10 – explain what means ASC-12 and HIT-6

Line 23 - explain in full name abbreviations MHD, MIDAS, and MAM

Introduction

Line 30 - explain in full name an abbreviation ICHD-3

Line 50 - explain in full name an abbreviation FDA

Line 59 - explain in full name an abbreviation e HER-MES

Results

Table 2 - remove the explanation of abbreviations from the table description that are not used in the table

explain in full name an abbreviation CFB

References

All literature items should be prepared in accordance with the format preferred by the journal. Please check and correct the literature.

Author Response

Dear reviewer, thank you very much for your suggestions which will surely be helpful in improving our work. 

Abstract: Line 9 – explain in full name an abbreviation CGRP; Line 10 – explain what means ASC-12 and HIT-6; Line 23 - explain in full name abbreviations MHD, MIDAS, and MAM.

We made all the corrections that you suggested.

Introduction: Line 30 - explain in full name an abbreviation ICHD-3, Line 50 - explain in full name an abbreviation FDA, Line 59 - explain in full name an abbreviation e HER-MES.

We made all the corrections that you suggested.

Results: Table 2 - remove the explanation of abbreviations from the table description that are not used in the table; explain in full name an abbreviation CFB

We made all the corrections that you suggested.

References: All literature items should be prepared in accordance with the format preferred by the journal. Please check and correct the literature.

About that point, we used a software to create the references list, and is not a problem to change it. However as stated in the Instructions for Authors on mdpi website "Your references may be in any style, provided that you use the consistent formatting throughout. It is essential to include author(s) name(s), journal or book title, article or chapter title (where required), year of publication, volume and issue (where appropriate) and pagination. DOI numbers (Digital Object Identifier) are not mandatory but highly encouraged. The bibliography software package EndNoteZoteroMendeleyReference Manager are recommended". So if there is a suggested style, please tell us. 

Regards

Reviewer 2 Report

Comments and Suggestions for Authors

The paper on the effectiveness of Anti-CGRP monoclonal antibodies compared to OnabotulinumtoxinA for chronic migraine treatment presents a comprehensive analysis but also has some limitations in its methodology:

1. Selection Bias: The inclusion of patients who had available follow-up data for HIT-6 or ASC-12 might lead to selection bias. Patients who return for follow-up and complete questionnaires might have different characteristics (more favorable result) compared to those who do not, potentially skewing the results.

2.
Lack of Randomization: Without random assignment of treatments to patients, there could be underlying differences between the groups that affect the outcomes. For instance, patients receiving BoNT-A had higher baseline monthly headache days, more comorbidities, and more frequent use of concomitant medications, indicating a possibly more severe migraine condition than those receiving Anti-CGRP mAbs.

3.
Generalizability: Conducted in two hospital centers in Italy, the study's findings might not be generalizable to all populations or healthcare settings. Differences in healthcare systems, patient demographics, and treatment protocols can affect the applicability of the results to other contexts.

  1. 4. Sample Size and Power: The study did not perform a formal sample size estimation for secondary outcomes, including HIT-6 and ASC-12, rendering the analyses exploratory. This limitation means the study may not have been adequately powered to detect differences in these outcomes, potentially affecting the reliability of the findings.

  2. 5. Subjective Outcome Measures
    : HIT-6 and ASC-12 are self-reported measures, which can introduce subjectivity into the results. Patients' perceptions of their symptoms and the impact on their quality of life can vary, affecting the consistency and reliability of these measures.

    The limitations mentioned above necessitate careful interpretation and suggest the need for further research,

Author Response

dear reviewer, thank you for your comments which will surely be helpful in improving our work. 

1. Selection Bias: The inclusion of patients who had available follow-up data for HIT-6 or ASC-12 might lead to selection bias. Patients who return for follow-up and complete questionnaires might have different characteristics (more favorable result) compared to those who do not, potentially skewing the results. R: The analysis reported here is a secondary outcome of the main study; the analysis was done on that cohort of patients whose follow-up was available for the parameters examined. Unfortunately, in clinical practice, many data can also be lost due to low patient compliance. This certainly constitutes a bias of a retrospective study in general. 

2. Lack of Randomization: Without random assignment of treatments to patients, there could be underlying differences between the groups that affect the outcomes. For instance, patients receiving BoNT-A had higher baseline monthly headache days, more comorbidities, and more frequent use of concomitant medications, indicating a possibly more severe migraine condition than those receiving Anti-CGRP mAbs. R:The study is a retrospective comparison study between two treatments, so randomization is infeasible. Due to the different characteristics between the two populations at baseline, an adjustment was made to the analyses conducted. 

3. Generalizability: Conducted in two hospital centers in Italy, the study's findings might not be generalizable to all populations or healthcare settings. Differences in healthcare systems, patient demographics, and treatment protocols can affect the applicability of the results to other contexts. R:The study is a multicenter, of two advanced level centers in italy, albeit in two separate regions offer similar levels of care to each other and similar to the entire Italian national health care system. So the observation has significance at least on a national level as a system of care, and internationally on a scientific level. The sample is not impressive, but it can be of cue for the complex migraine population afferent to headache centers. 

4. Sample Size and Power: The study did not perform a formal sample size estimation for secondary outcomes, including HIT-6 and ASC-12, rendering the analyses exploratory. This limitation means the study may not have been adequately powered to detect differences in these outcomes, potentially affecting the reliability of the findings. R: The power study is done on the primary endpoint and so it was done for the main study; the analysis presented here is an exploratory analysis on a secondary outcome.

5. Subjective Outcome Measures: HIT-6 and ASC-12 are self-reported measures, which can introduce subjectivity into the results. Patients' perceptions of their symptoms and the impact on their quality of life can vary, affecting the consistency and reliability of these measures. R: Like all PROMs, an individual subjectivity comes into play. In the absence of objectifiable data of routine use in the clinic, these are the best secondary indicators of clinical outcomes that we can benefit from.